# MXene Core-Shell Nanosheets: Facile Synthesis, Optical Properties, and Versatile Photonics Applications

**DOI:** 10.3390/nano11081995

**Published:** 2021-08-03

**Authors:** Yunjia Wang, Shunxiang Liu, Feng Zhu, Yiyu Gan, Qiao Wen

**Affiliations:** Key Laboratory of Optoelectronic Devices and Systems of Ministry of Education and Guangdong Province, College of Physics and Optoelectronic Engineering, Shenzhen University, Shenzhen 518060, China; wangyunjia@szu.edu.cn (Y.W.); 2150190124@email.szu.edu.cn (S.L.); 1160713978@szu.edu.cn (F.Z.); 2151190226@email.szu.edu.cn (Y.G.)

**Keywords:** 2D material, core-shell structure, mode-locked, single frequency, fiber laser

## Abstract

In recent years, the transition metal carbonitrides(MXenes) have been widely applied to photoelectric field, and better performance of these applications was achieved via MXene complex structures. In our work, we proposed a MXene core-shell nanosheet composed of a Ti_2_C (MXene) phase and gold nanoparticles, and applied it to mode-locked and single-frequency fiber laser applications. The optoelectronic results suggested that the performances of these two applications were both improved when MXene core-shell nanosheets were applied. As a result, we obtained a mode-locking operation with 670 fs pulses, and the threshold pump power reached to as low as 20 mW. Besides, a single-frequency laser with the narrowest linewidth of ~1 kHz is also demonstrated experimentally. Our research work proved that MXene core-shell nanosheets could be used as saturable absorbers (SAs) to promote versatile photonic applications.

## 1. Introduction

Over the past decade, researches on two-dimensional (2D) materials [1] have drawn much attention, and these 2D materials included transition metal dichalcogenides (TMDs) [2], semiconductors [3,4], and insulators [5]. Due to the strong interaction with lightwaves, more and more applied fundamental researches of 2D materials have concentrated in versatile photonic applications [6]. Nonlinear optical characteristic was one of the most important properties of 2D materials. By taking full advantage of the unique nonlinear optical characteristic, 2D materials have been widely applied to laser technology, laser processing [7], laser displays [8], photodetectors [9], laser facilities [10], and medical treatment [11]. Graphene, as the earliest discovered 2D materials, possesses a zero band gap and has been proved to achieve good nonlinear optical characteristics. But the low absorption coefficient and low damage threshold have limited its development [12]. BP, as a tunable band gap semiconductor, has been demonstrated to exhibit remarkable anisotropy, high carrier mobility, and good stability with a layered structure [13]. However, when it was applied to photonics researches, the exfoliation of few-layer nanosheet structure, seriously decreased the stability of BP and further restricted its application in photonics.

To our best knowledge, MXenes, as one of the novel 2D materials, have been reported to be achieved from the MAX(M = transition metal, A = Al, X = C,N) phase. Because of the treatment by HF solution, the surface of MXenes is rich in functional groups [14], such as -F, -O, and -OH group. Such surface groups have promised MXene materials a flexible surface modification [15]. Moreover, the layered structure has ensured MXene materials tunable properties [16,17], and made them more potential in photoelectronics [18], photocatalysis [19], and energy storage [20]. Our team has demonstrated the nonlinear optical characteristics of MXenes in erbium-doped fiber (EDF) lasers. Based on the functional groups of MXene materials and our previous work, we have further promoted the performance of MXenes by fully utilizing the advantages of MXene@gold core-shell nanosheets.

As we known, MXene core-shell nanostructures usually consist of MXene materials and other materials via the effect of functional groups. By applying core-shell nanostructures, the photoelectric properties of MXene materials were demonstrated to be modified, such as absorption enhancement or shiftness of the absorption peak [21]. If MXene materials are enclosed by nanometal materials, or by other materials that possess collective free electrons, such as gold nanoparticles or silver nanoparticles, then the optical characteristics are expected to be enhanced due to surface plasmon effect [22]. For example, MXene@gold core-shell nanocomposites have been reported to exhibit highly efficient catalysis ability, and effectively enhanced the second biological window [23,24]. These studies suggested that the strong photoelectronic interaction is fully utilized by MXene@gold core-shell structure, and excellent performance is achieved. However, the surface plasmon effect of metal nanoparticles on MXene materials has not been deeply demonstrated, application of the MXene@gold nanostructure to promote the performance in photonics technology has seldom been reported, and its improvement mechanism has not yet been explored [25]. 

To our best knowledge, ultrafast fiber lasers (UFFLs) and single-frequency fiber lasers (SFFLs) are two important fiber lasers in our daily life and industrial manufacturing. The UFFL with an ultrashort pulse width is a significant component in fiber communications [26], micromachining [27], and medical surgery [28]. Hence, researchers have devoted themselves to improving the characteristics of UFFLs, which included a lower lasing threshold, an ultrashort pulse width, a high peak power, and a high repetition frequency. In addition, a stable SFFL with narrow linewidth and low noise is an ideal candidate for coherent optical communications [29], precise optical sensing [30], gravitational wave detection [31], and lidar [32]. It should be noted that, the narrow linewidth of SFFL has been eagerly pursued, and the reason was maily ascribed to the urgent need of high-precision sensing systems [33]. Recently, Ning Xu et al. have synthesized MXene (Ti_3_C_2_Tx) quantum dots from few-layer MXene materials, and applied it to UFFLs and SFFLs [34]. The results suggested that better nonlinear optics was achieved due to quantum effect, and indicated that the laser characteristics can be optimized by improving the optical nonlinear response of MXenes materials.

In our work, Ti_2_C@Au core-shell nanosheets were prepared and employed as a saturable absorber (SA) to construct an UFFL and a SFFL. Compared with the UFFL and SFFL, which employed Ti_2_C nanosheets as a SA [35], the laser characteristics of both UFFL and SFFL were conspicuously promoted. The results suggested that the UFFL based on Ti_2_C@Au core-shell nanostructure has a lasing threshold as low as 20 mW, and the SFFL in this work has a laser linewidth as narrow as 1 kHz, which were the best performances among the relevent lasers based on MXenes. It indicated that the Ti_2_C@Au core-shell nanostructure was a potential material for achieving low threshold, low noise, and narrow linewidth in EDF-SFFLs. Our study provided insight into promoting MXene materials in fiber laser applications.

## 2. Results and Discussions

### 2.1. Fabrication and Characterization of Ti_2_C@Au

In Brief, Ti_2_C@Au nanostructures were synthesized from Ti_2_C materials, which were fabricated from the Ti_2_AlC phase. First, the purchased Ti_2_AlC solid phase was dissolved in HF solution, and then ultrasonicated to achieve a few-layer Ti_2_C solution. Second, the Ti_2_C solution was centrifugated at 3000 r/min for 2 min to obtain small Ti_2_C particles. Subsequently, HAuCl_4_ crystals were dissolved in deionized water and then combined with the Ti_2_C particle dispersion in a beaker. The redox reaction of HAuCl_4_ and Ti_2_C particles was conducted at 40 °C for 30 min to prepare core-shell nanostructures. After the reaction process, the composites were centrifuged and dispersed in an isopropanol solution before use.

The flowchart in Figure 1 shows the preparation process of Ti_2_C@Au core-shell nanostructure. The reaction between MXene materials and Au^3+^ can be expressed by the following equation:
(1)
$${\mathrm{Ti}}_{2}{\mathrm{CF}}_{x}{\left(\mathrm{OH}\right)}_{y}+3{\delta e}^{-}+{\delta \mathrm{Au}}^{3+}\to {\mathrm{Ti}}_{2}{\mathrm{CF}}_{x}{\left(\mathrm{OH}\right)}_{y}+\delta \mathrm{Au}$$


In Equation (1), Ti_2_CF_x_(OH)_y_ represented Ti_2_C (MXene) materials, whose surface showed -F and -OH groups, HAuCl_4_ crystal could be dissolved into Au^3+^ in solution. During the reaction, a mounts of Au^3+^ were reduced to gold nanoparticle and deposited onto Ti_2_C (MXene) materials when reacted with Ti_2_CF_x_(OH)_y_. In our preparation experiment, the solvent concentration and the usage was strictly kept the same. The concentration of the Ti_2_C material and the Ti_2_C@Au core-shell nanostructure, were both adjusted to 3 mg/mL with isopropanol.

Figure 2 shows the scanning electronic microscopy (SEM) characterization and transimission electronic microscopy (TEM) characterization of the prepared Ti_2_C material and Ti_2_C@Au core-shell nanostructure. 

Figure 2a shows the SEM image of Ti_2_C material, which exhibited a morphology of layer structure, while Figure 2b shows the SEM image of Ti_2_C@ Au core-shell nanostructure, which presented densitive spots deposited on the Ti_2_C materials. Figure 2c–f shows the TEM images of the prepared Ti_2_C materials and the Ti_2_C@Au core-shell nanostructure, to further analyze the transformation from the Ti_2_C material to the Ti_2_C@Au nanostructure. As depicted, a core-shell-like nanostructure with a shell thickness of approximately 15 nm was found, and the crystal lattice distance was enlarged from 0.42 to 0.58 nm when the Ti_2_C material was transformed into the Ti_2_C@Au core-shell nanostructure. 

Figure 3 shows the X-ray photoelectronic spectrum (XPS) of Ti_2_C@Au core-shell nanostructure for further analyzing. As depicted, the binding energy intensity was enhanced when the spots deposited on the Ti_2_C material. From Figure 3d, we confirmed the spots to be gold nanoparticles with the binding energies of 83 eV and 87 eV. In addition, the titanium elemental analysis exhibited in Figure 3b, shows the binding energy intensity of Ti_2_C material is remarkly enhanced with densitive coverage of gold nanoparticles. A similar phenomenon was found for carbon element, which is shown in Figure 3c. These results indicated that the outer environment of Ti_2_C materials was modified when covered by gold nanoparticles [36,37,38]. 

Figure 4a shows the absorption spectra of Ti_2_C materials and Ti_2_C @Au nanostructure. As depicted, Ti_2_C material and Ti_2_C@Au nanostructure both exhibited a broad absorption range from 400 to 2000 nm. It should be noted that there were six optical peaks appeared in the absorption spectra of Ti_2_C materials, which were respectively located around 900, 1015, 1470, 1590, and 1710 nm. It indicated that abundant energy level was achieved for the prepared Ti_2_C material, however, when the gold nanoparticles deposited on Ti_2_C material, the absorption peak of 1710 nm was significantly enhanced. To our best knowledge, gold nanoparticles hardly reacted with the Ti_2_C materials via chemical interaction, so we establish that the absorption enhancement originated from the physical interaction when excited by an incident with wavelength around 1710 nm. Owing to the collective oscillation of free electrons, electromagnetic field that local around gold nanoparticles was considered to be formed, and attributed to the absorption enhancement of Ti_2_C material.

As we know, the Raman scattering signals of most molecules could be enhanced due to the electromagnetic field excitation. So we applied the Raman spectrum to demonstrate the field-enhancemnet effect of Ti_2_C@Au nanostructure, whose result is shown in Figure 4b; Ti_2_C materials was measured for a comparison. As depicted, we infered the characterized peak that located around 1050 cm^−1^ originated from the vibration of carbon atom, and another peak that exhibited around 1600 cm^−1^ was due to the vibration of –OH group. However, when the gold nanoparticles deposited onto Ti_2_C material, significant enhancement was observed of such characterized peaks. These results suggested that photoelectric interaction was fully utilized when gold nanoparticle and Ti_2_C material were assembled together in such a complex structure, and attributed to the field enhancement effect.

### 2.2. Optical Nonlinearity Characteristics

Figure 5 shows a balanced twin-detector measurement system to study the nonlinear optical characteristics of the fabricated Ti_2_C@Au-SA. The seed source is a homemade erbium-doped fiber laser with a pulse duration of 600 fs and a repetition rate of 8.13 MHz. After the measurement, we found that the fabricated Ti_2_C@Au-SA has a modulation depth of 6.6%, a saturation intensity of 17.96 GW/cm^2^, and the corresponding non-saturable loss of 18.9%.

### 2.3. Mode-Locked Fiber Laser

To analyze the performance of Ti_2_C @Au in erbium-doped fiber laser, the Ti_2_C @Au saturable absorbers are transferred to an all-fiber laser setup, and pumped by a 980 nm laser diode with a maximum power of 750 mW via a wavelength-division multiplexer, as shown in the Figure 6.

The characteristics of the stable femtosecond pulse results are shown in Figure 7a–d. As depicted, the stable mode-locked regime was obtained when the pump power was larger than 20 mW, while, to our best knowledge, the obtained mode-locking threshold is the lowest among the MXenes at a wavelength of 1.55 um (see Appendix A). Figure 7a displays the mode-locked pulse train. The time interval between the two pulses was 117.6 ns, which corresponded to 8.5 MHz repetition rate. Figure 7b shows the optical spectrum was centered at 1560 nm with mode-locked pulses and a 3-dB spectral width of 3.2 nm.

Figure 7c presents a strong signal peak with a mode-locked repetition rate of 8.5 MHz, and the signal to noise ratio (SNR) of about 62 dB. The autocorrelation trace of corresponding mode-locked pulse is shown in Figure 7d. When the sech^2^ function was used to fit the measured pulse autocorrelation trace, the estimated pulse duration was 670 fs with a time-bandwidth product (TBP) of 0.54. Figure 7e presents the relationship between average output power and pump power, and it shows an excellent linear relationship with a slope efficiency of 3.08%. In our experiment, the laser was operated in continuous wave (CW) regime when the pump power was increased to 5 mW, as the pump power further increased to the threshold of 20 mW, stable continuous wave mode lock (CWML) was achieved. Figure 7f shows the pulse spectrum that applied the same material solution evolved over 10 h, which indicated that higher stability of the Erbium-doped femtosecond fiber laser was obtained. For a convenient comparison, we listed the properties of EDFs based on 2D materials in Appendix A, which was shown in the Appendix A.

To explain the better performance for mode-locked fiber laser when applied the Ti_2_C@Au SA, we owned the reason to the surface plasmon effect that originated from the gold nanoparticles, which resulted in the enhancement of nonlinear optics [39]. 

Figure 8 shows the schematic diagram illustration of the mode-locking laser by applied Ti_2_C@Au core-shell nanostructures. As depicted, a mount of gold nanoparticles deposited on the surface of Ti_2_C materials, and excited by laser source with wavelength of 1500 nm, then collective electrons are aroused and form oscillation around the interface between gold nanoparticle and Ti_2_C materials, resulting in local electromagnetic field enhancement [40]. Under the affection of electromagnetic filed, the electronic cloud of MXene core-shell nanostructures is distorted, and the Ti_2_C material molecules trend to reorient and lead to a change of electronic density, as well as the variation of refractivity [41]. Thus, the strong light wave that coupled into fiber laser, can be effectively modulated via the interaction with Ti_2_C@Au core-shell nanostructures. That is, the photonic mode which is harmonic with the polarization direction, is effectively enhanced, while those that are not harmonic with the polarization direction weaken [42].

To our best knowledge, nonlinear absorption optic of dielectric, is correlated with the modification of refractive index, which is proportional to the square of electric field.

Here, the nonlinear absorption coefficient of Ti_2_C@Au nanostructures that applied to fiber laser is given by [43]: 
(2)
$$\alpha_{\mathrm{non},G}=\frac{\alpha_{0}}{1+I_{e,t}/I_{e,\mathrm{sat}}}\to I_{e,t}=I_{e,\mathrm{sat}}(\frac{\alpha_{0,G}}{\alpha_{\mathrm{non},G}}-1)$$

where 
$\alpha_{0,G}$
 is the nonlinear absorption component, 
$I_{e,t}$
 represents the evanescent-wave intensity, and 
$I_{e,\mathrm{sat}}$
 represents the saturable intensity of Ti_2_C@Au nanostructures. The evanescent-wave within the hole-cladding region of the fiber laser, was interfered by the evanescent-wave that originated from the gold nanoparticles, and the evanescent-wave intensity exponentially decayed with the radial distance (x) away from the core/cladding interface. We described the evanescent-wave intensity by following equation [44]: 
(3)
$$I_{e,\mathrm{sat}}=I_{0}e^{-2\beta x}$$

where 
$I_{0}$
 and 
β
 are respectively the pulse intensity of intracavity and the extinction coefficient factor. With the assistance of Ti_2_C@Au nanostructures, the extinction coefficient factor is expressed by Equation (4) [45]:
(4)
$$\beta =\frac{2{\pi n}_{c}}{\lambda }{\left(\frac{\sin^{2}\theta }{{\left(\frac{n_{c}}{n_{i}}\right)}^{2}}-1\right)}^{2}$$

where 
$n_{i}$
 is the core refractive index, and 
$n_{c}$
 is the effective cladding refractive index. In our work, the core is acted by Ti_2_C materials, and the cladding are gold nanoparticles. To our best knowledge, gold nanoparticle possesses a higher refractive index than Ti_2_C materials, so the value of 
$\frac{n_{c}}{n_{i}}$
 is larger than 1, and the extinction coefficient factor 
β
 is negative. According to Equations (2) and (3), the 
$I_{e,\mathrm{sat}}$
 and 
$\alpha_{0,G}$
 are positively increased compared to the pure MXene materials, which coincided with the P-scan measurement of the Ti_2_C@Au nanostructures. It indicated that the nonlinear absorption is effectively improved by gold nanoparticles, and contributed to the lower threshold output of ultrashort pulse.

### 2.4. Single Frequency Fiber Laser

To further understand the nonlinear optic enhancement effect by using Ti_2_C@Au core-shell nanostructures, the Ti_2_C@Au was applied as a SA to single frequency fiber laser (SFFL). Figure 9 shows the experimental schematic of this SFFL which included a seed laser and an amplifier.

Pump 1 was launched into the laser cavity through a fiber bragg grating (FBG, reflection of 57% at 1549.65 nm, 3 dB bandwidth of 0.036 nm) via a 1 × 2 wavelength division multiplexer (WDM, 980/1550 nm). A length of 0.3 m single mode EDF (peak absorption of 17 dB/m at 1530 nm) was severed as the gain medium. In purpose of eliminating the effect on the loop mirror by the residual pump, another WDM (1 × 2.980/1550 nm) was spliced to the EDF. On the end of the cavity, the loop mirror that connected the WDM by an optical coupler(OC, 50/50), was worked as a high reflectivity mirror to bring laser beam back. A section of tapered fiber, with a length of ~3 cm and a waist diameter of ~10 μm, was embedded into the loop mirror to release the optical evanescent filed. To optimize the reflection of the loop mirror, a polarization controller (PC2 and PC3) was set to adjust the polarization of the counter-propagation beam. As we know, the spatial hole buring (SHB) effect holds when two laser beams are encountered in the cavity. Thus the PC1, which was between FBG and gain medium, was employed to restrain this effect [46]. It should be noted that the total length of the cavity was ~2 m, it was not easy to detect and analyze the laser signal due to the low power of seed laser. Then, the seed laser was incidented into an amplifier, composed of a pump source (Pump 2, 980 nm) and a certain length EDF. Finally, the amplified laser beam propagated along a Fabry-Perot Scanning Interferometer (FPSI, Thorlabs, SA200-12B, free spectral range of 1.5 GHz, resolution of 7.5 MHz) through a pigtail fiber for inspecting the single-frequency characteristics. Besides, the linewidth was investigated by a delayed self-homodyne (DSH) technique, which has a delay line of 50 km and a frequency shifter of 200 MHz. The signal was detected by a high-speed photodetector (EOT, ET-5000F), and presented in a radio frequency spectrum analyzer (RFSA, Rigol, DSA815). 

To keep the laser in a peak power output, the power of Pump 1 and Pump 2 were both set at a appropriate value, and the two PCs in the loop were also adjusted. Before MXene core-shell nanostructure solution was dripped over the taper fiber, the FPSI recorded the scanning spectrum as shown in Figure 10 by pink curve. This result indicated that the laser was operated at a multi-frequencies state, and never disappeared even though the three PCs were adjusted. Figure 11 presents the laser beam gets through the DSH system with pink curve, whose heterodyne output signal of the laser was obtained from nonsolution state. As depicted, a number of unstable peaks can be observed in the range of 0~400 MHz, which was due to the multi-frequencies. As the Ti_2_C@Au solution was added to the tapered fiber, the counter propagating beam created a standing wave, and established an instantaneous grating in the intra-loop simultaneously [47]. After a while, the signal on scanning spectrum turned into the blue curve, which is shown in Figure 10. Despite the multi-frequencies state was unaltered, the number of frequencies was decreased obviously. It can be imagined that the amount of frequencies measured by DSH will decline as well.

Figure 11 validates our prediction by the blue curve. We insisted that this phenomenon was induced by the SHB effect. To elimate the SHB effect, we carefully rotated the PC1, and the reflection of loop mirror was also optimized by adjusting PC2 and PC3 patiently. As a result, a typical scanning spectrum of SFFLs was achieved, which is presented in Figure 10 by the red curve. It suggested that the laser was just operated at a single frequency state, and a single peak centered at ~200 MHz appeared in the RFSA, which is shown in Figure 11 by the red curve.

For investigating the linewidth of this SFFL, the frequency signal ranging from 198 to 202 MHz was obtained by the DSH technique and the RFSA with a resolution of 600 Hz, then fitted by Lorentzian curve, whose result is shown in Figure 12a. It is suggested that the bandwidth of the peak at −20 dB was 20.63 kHz by the fitted curve, whose result coincided with a laser with a linewidth of 1.0315 kHz.

Afterward, we adjusted two pump power to research the relationship between pump power and output power. First, the Pump1 power was kept under 85 mW until the laser stops operating. Then, the frequency signal of the laser gets unstable, and the mode became conspicuous when Pump 1 power was higher than 500 mW. Second, we fixed the Pump1 power at 500 mW for the highest output power with single frequency operation, and the Pump 2 power was also altered from 50 to 700 mW. Figure 12b shows the output power of seed laser versus pump1 power, as well as the output power of amplified laser versus pump 2 power, and the results were fitted by line as well. A slop efficiency of ~0.207% for the seed laser, and ~5.64% for the amplified laser were calaculated. We believed that the short gain medium and the unsaturable loss of materials, contributed to the low slop efficiency. An optical spectrum analyzer (OSA, YOKOGAWA AQ6370D) was employed to analyze the SNR, which reached to the highest value when the Pump 2 power was locked at 300 mW. Figure 12c exhibits that the SNR exceeded 48 dB. In this situation, the output power of seed laser and amplified laser were recorded every two minutes within 1 h, and the results are shown in Figure 12d. It indicated that the power of seed laser, as well as the amplified laser, had a fluctuation in the range of 1.179 ± 0.009 mW and 11.17 ± 0.15 mW, showing that the results support the stability of less than 0.76% and 1.34% respectively. In addition, the average powers were calculated to be 1.179 mW and 11.160 mW, the stability of laser power was evaluated by the standard error (SE), with values ~8.152 × 10^−4^ and 1.496 × 10^−2^ respectively. As far as we are concerned, the increasing of power and the amplified spontaneous emission (ASE), should be accountable for the deteriorated output power stability of amplified laser.

For a convenient comparision, we listed the properties of SFFLs based on LCLM with 2D material in Appendix A, it can be found that our work was reported to be the narrowest linewidth among this kind of SFFL, whose comparison is shown in Appendix A.

## 3. Conclusions

In conclusion, we successfully synthesized a core-shell nanostructure composed of Ti_2_C (MXene) phase and gold nanoparticle. The morphology and composition were characterized by SEM, TEM, and XPS, the optoelectronic characteristic was analyzed by UV-vis spectrum and Raman spectra. The results suggested that full utilization of surface plasmon effect was made by gold nanoparticle, and attributed to the application of nonlinear optics. Finally, we applied such core-shell nanostructure to a passive mode-locked EDFL and an EDF-SFFL. The results suggested that stable passive mode-locked EDFL has a low startup threshold, and approximately 410% lower than the previously reported. The pulse duration and the center wavelength of this laser were measured to be 670 fs and 1560 nm respectively. The stabilized EDF-SFFL based on a linear cavity with a loop mirror embedded in a taper fiber, has a linewidth of ~1.00 kHz when immersed in such Ti_2_C@Au core-shell material. To our best knowledge, this was the narrowest linewidth of the SFFL based on 2D materials. It has a SNR of 31.82 dB, and an average output power of ~1.18 mW and ~11.16 mW before and after amplification respectively. While, the output power fluctuation of seed laser and amplified laser was less than 0.76% and 1.34% respectively. It proved that the Ti_2_C@Au core-shell material can be employed to suppress the bandwidth and enhance the stability of SFFL. We believe that the passive MLFL and SFFL can be improved further by optimizing the Ti_2_C@Au core-shell material.

## Figures and Tables

**Figure 1 nanomaterials-11-01995-f001:**
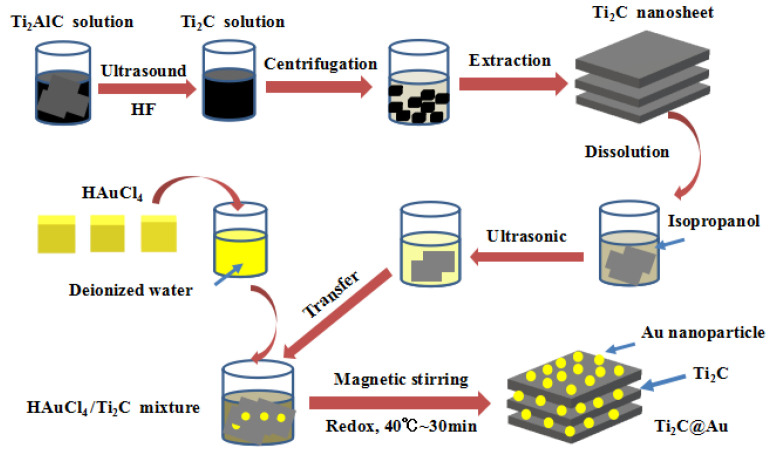
Schematic illustration of the Ti_2_C@Au nanostructure preparation.

**Figure 2 nanomaterials-11-01995-f002:**
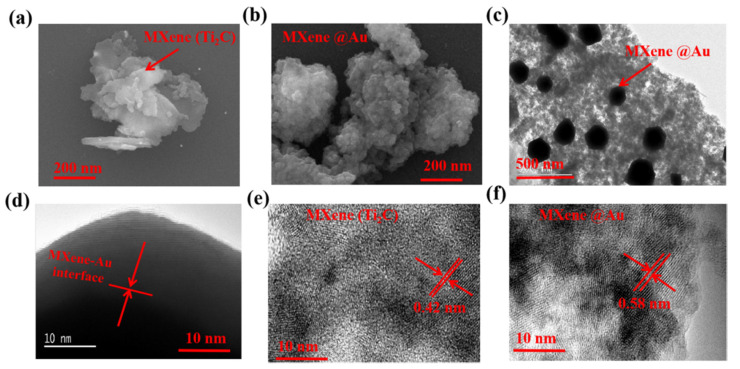
(**a**) SEM image of Ti_2_C nanostructure; (**b**) SEM image of Ti_2_C@Au nanostructure; (**c**,**d**)TEM image of Ti_2_C @Au nanostructure; (**e**,**f**)Ti_2_C nanostructure and Ti_2_C@Au nanostructure analyzed by resolution TEM.

**Figure 3 nanomaterials-11-01995-f003:**
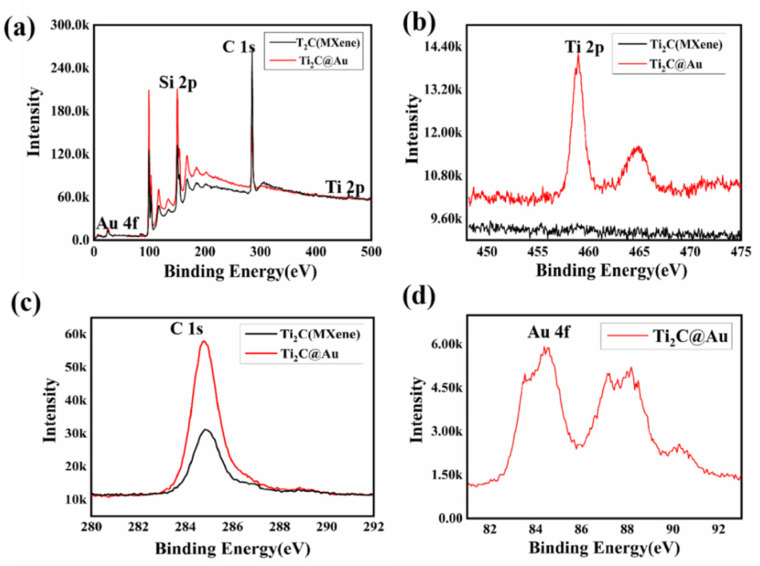
(**a**) XPS analysis of the Ti_2_C material and Ti_2_C@Au nanostructure; (**b**) titanium elemental analysis; (**c**) carbon elemental analysis; (**d**) gold elemental analysis.

**Figure 4 nanomaterials-11-01995-f004:**
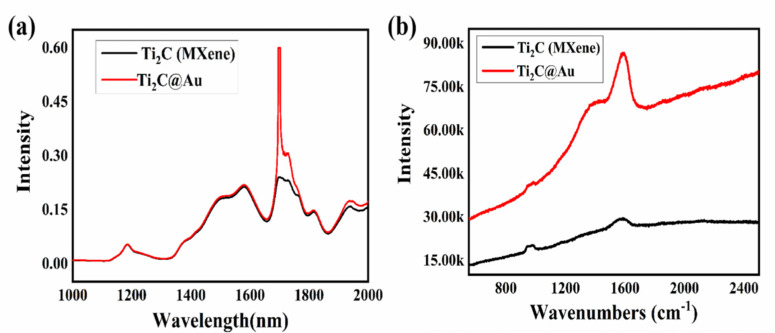
(**a**) Absorption spectra of equal amounts of Ti_2_C materials and Ti_2_C@Au core-shell nanostructure; (**b**) Raman spectrum of Ti_2_C materials and Ti_2_C@Au core-shell nanostructure.

**Figure 5 nanomaterials-11-01995-f005:**
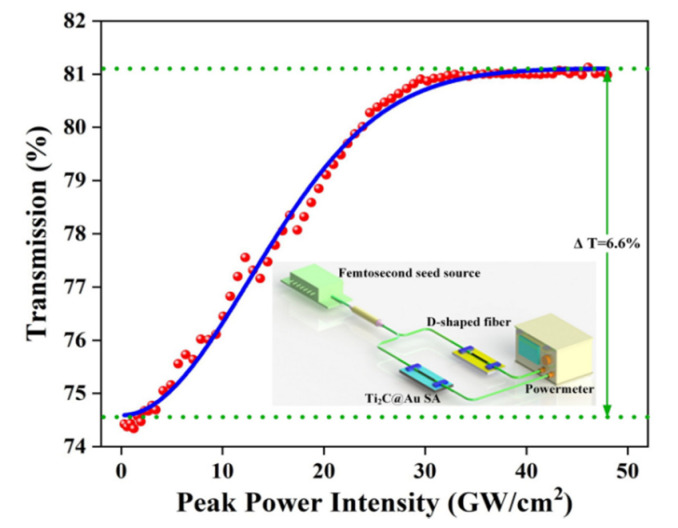
Nonlinear transmittance of the Ti_2_C@Au-SA as a function of peak power intensity.

**Figure 6 nanomaterials-11-01995-f006:**
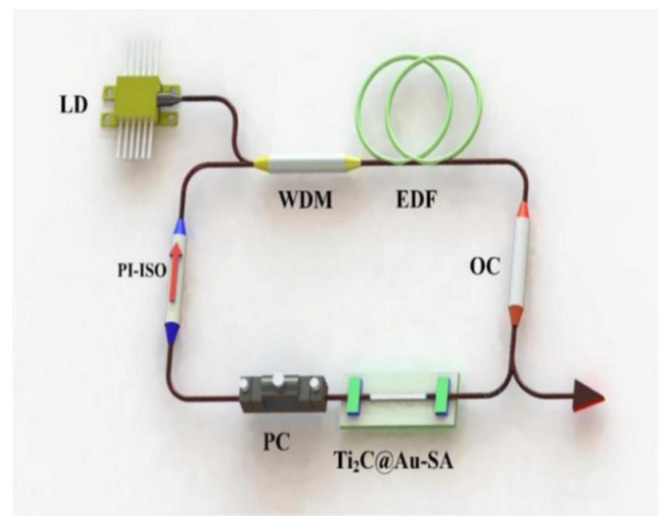
Configuration of the pulsed laser or ultra-short pulsed laser based on Ti_2_C @Au. LD: laser diode. WDM: wavelength division multiplexer. EDF: erbium-doped fiber. OC: output coupler. Ti_2_C @Au-SA: Ti_2_C @Au saturable absorber. PC: polarization controller. PI-ISO: polarization independent isolator.

**Figure 7 nanomaterials-11-01995-f007:**
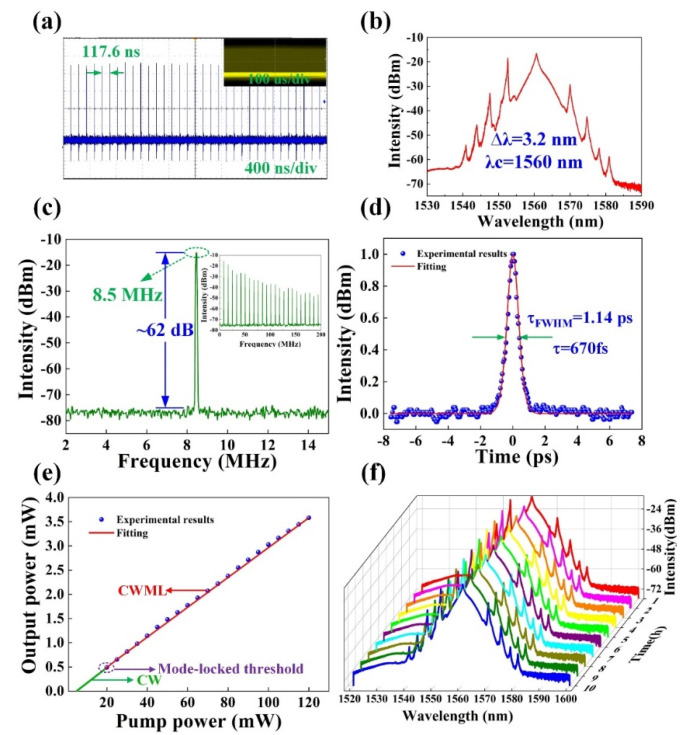
Typical mode-locking characteristics. (**a**) Pulse train. (**b**) Optical spectrum. (**c**) Radio frequency spectrum (inset: the wideband RF spectrum). (**d**) Autocorrelation trace. (**e**) Variation of the output power with respect to the pump power. (**f**) Optical spectra measurements at 1 h intervals over 10 h.

**Figure 8 nanomaterials-11-01995-f008:**
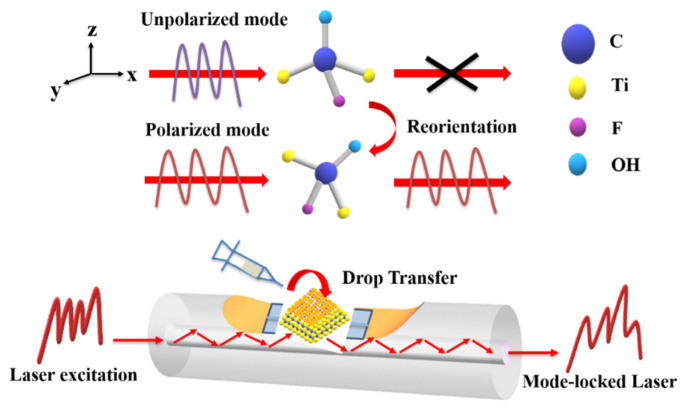
Schematic illustration of mode-locked fiber laser by using MXene core-shell nanostructures.

**Figure 9 nanomaterials-11-01995-f009:**
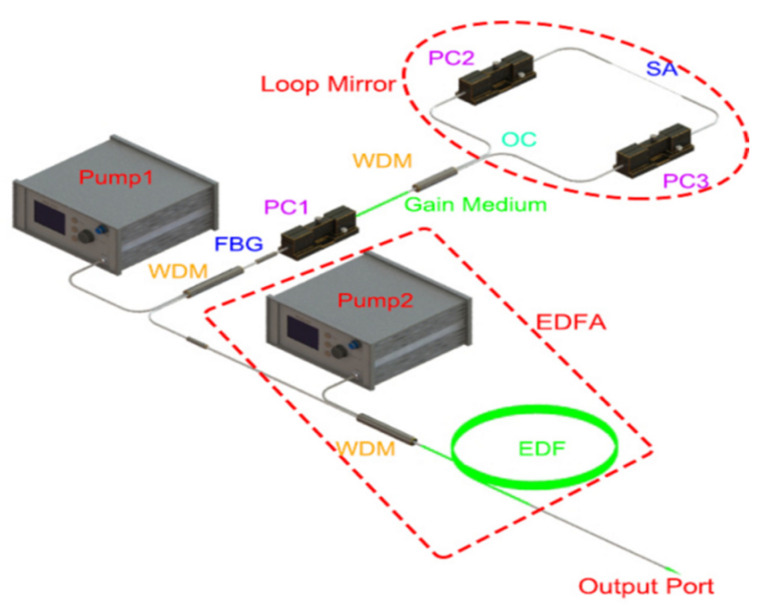
Experimental schematic of SFFL based on LCLM embedded a SA.

**Figure 10 nanomaterials-11-01995-f010:**
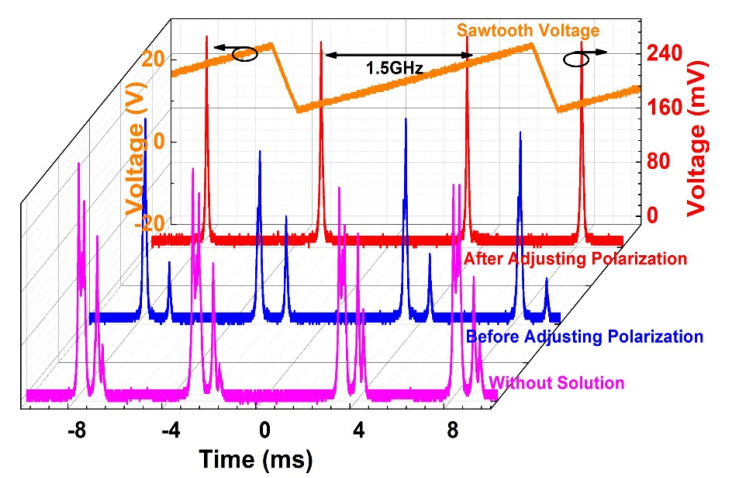
Evolution of FPSI scanning spectrum.

**Figure 11 nanomaterials-11-01995-f011:**
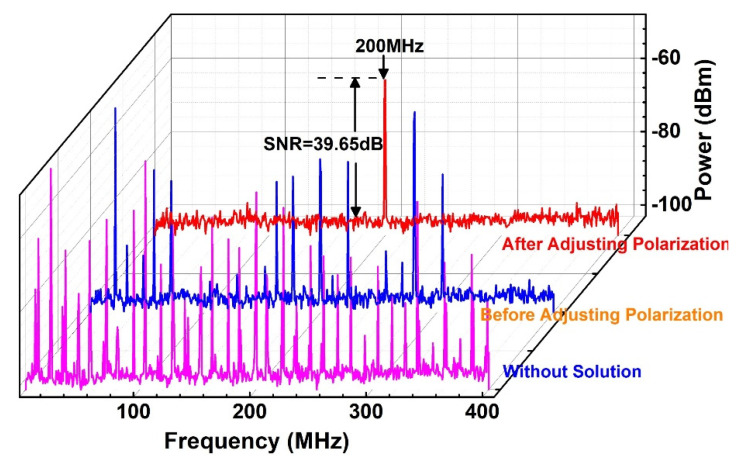
Evolution of FR scanning spectrum.

**Figure 12 nanomaterials-11-01995-f012:**
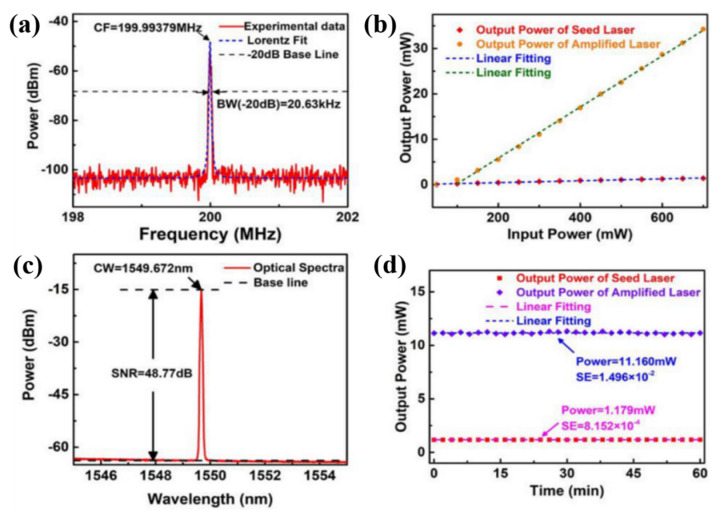
The spectrum measured by DSH technique and the Lorentz fit (**a**); the optical spectrum of amplified laser (**b**); output power of seed laser and amplified laser versus to input pump power (**c**); the output power stability of seed laser and amplified laser in 1 h (**d**).

## Data Availability

Data underlying the results presented in this paper are not publicly available at this time but may be obtained from the authors upon reasonable request.

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
