# Peer review of "MXene Core-Shell Nanosheets: Facile Synthesis, Optical Properties, and Versatile Photonics Applications"

_nanomaterials, 2021, doi:10.3390/nano11081995_

Round 1

Reviewer 1 Report

I think it is a very complete work,  the introduction include all relevant references, and all the work is completed with a lot of references, the research is very complete, the conclusion are supported by all results very well explained.

Author Response

   Thank you for your positive evaluation of our work! To further improve our manuscript, we have made some revisions in our revised manuscript, please see the attachment!Thank you!

Reviewer 2 Report

The manuscript is not in the reviewable stage. Extensive grammatical errors are present everywhere and these render this manuscript not suitable for review in the present format. These errors exist almost everywhere:  in the abstract, introduction, results, and conclusion sections. It also includes undefined acronyms such as SA, etc., which makes the job of the reader difficult. The authors should improve the quality of the Figures/visibility as well, such as improved font sizes, labels, and legends, and cite borrowed equations if have any.  The research work is interesting however, I cannot review this manuscript further unless the authors revise and resubmit it.  

Author Response

   Thank you for your valuable comments! We have tried our best to revise our manuscript according to the suggestions, and the revision details are shown in the following attachments. Please see the attachments, thank you!

Reviewer 3 Report

Dear authors,  

This is an interesting manuscript for the Nanomaterials audience. English should be improved in various instances since it is hard to read and understand the significance of this work. Turnit-in showed a 24% similarity index, especially in paragraphs:2.1-2.4 with text from previous manuscripts, that should be revised. 

Specific Comments

P1, L14: ‘and a thresh- old pump power as low as 20 mW is achieved, and a single-frequency laser’ Please revise. 

P1, L17: ‘SAs’ Please define every abbreviation the first time cited in the text.  

P1, L22: ‘In recently,’ Please revise. 

P1, L28: ‘and medical treatment[11]’ I cannot find Ref 11 in the Journal’s site. Could authors provide it in supplementary materials in order to read the medical applications? 

P1, L40: ‘a tunable properties’ Please revise. 

P2, L46: ‘materials enclosed in another material or of other materials’ Please revise. 

P3, L18: Please explain the terms in equation 1. 

P4, L132: ‘furher’ Please correct. 

P4, L149: ‘was decorated’ Is this the appropriate term? 

P5, L162: ‘when excited by a certain incident’ Please be more specific. 

P7, L229: ‘we owned it to the surface plasmon effect of the gold nanoparticles when excited by laser source and raised the electromagnetic filed,’ Please revise and correct. 

P10, L345: ‘Then we did a research for the impact on this laser by changing two pump power.’ Please revise. 

P10, L357: ‘Since we fixed the Pump 1 power at 500 mW and altered the Pump 2 power from 50 mW to 700 mW’ Please explain. 

Author Response

    Thank you for your valuable comments! We have revised our manuscript point by point according to the comments! Please see the attachment!

Round 2

Reviewer 2 Report

The authors have attempted to improve the quality of the manuscript although they did not mention how they did. Serious grammatical errors are still present in the text. See for example; Abstract "abtained" vs "obtained." The figure quality has been improved. Careful proofreading and correction might still be needed before sending it out for publication. 

Author Response

Thank you for your comments! We feel sorry that we have no mention where we have revised in our manuscript in the first round, however, we are pleased to further revised our manuscript by careful proofreading and grammar correction before being sent to publication. For example, we have corrected the word “abtained” into “obtained” in the Abstract section of our manuscript! Besides, we have added a word “most” at Page 1, Line 32 in our revised manuscript, some changes have also been made at Page 2, Line 68. Other grammar corrections have been made and highlighted with red color in our revised manuscript, please check, thank you! 

Reviewer 3 Report

Dear Authors

You have responded to all the comments thoroughly, so I have opted to recommend the Acceptance of the manuscript.  

Author Response

    Thank you for your positive evaluation of our manuscript! To further improve the quality of our manuscript, we have carefully checked the grammar again in our manuscript before it has been accepted for publication!
